



# RISK MANAGEMENT FRAMEWORK OF ENVIRONMENTAL HAZARDS AND EXTREMES IN MEDITERRANEAN ECOSYSTEMS

Panagiotis T. Nastos[1], Nicolas R. Dalezios[2], Ioannis N. Faraslis[3], Kostas Mitrakopoulos[2], Anna Blanta[2], Marios Spiliotopoulos[2], Stavros Sakellariou[2], Pantelis Sidiropoulos[2] and Ana María Tarquis[4]

[1]Laboratory of Climatology and Atmospheric Environment, National and Kappodistrian University of Athens, Athens, Greece (e-mail: nastos@geol.uoa.gr)
[2]Department of Civil Engineering, University of Thessaly, Volos, Greece (e-mail: dalezios.n.r@gmail.com)
[3]Department of Environment, University of Thessaly, Volos, Greece (e-mail: faraslis@uth.gr)
[4]CEIGRAM-Department of Agricultural Engineering, Technical University of Madrid, Madrid, Spain
 (e-mail: anamaria.tarquis@upm.es)

*Correspondence to*: Prof. Panagiotis Nastos (e-mail: nastos@geol.uoa.gr)

**Abstract.**

Risk assessment constitutes the first part within the risk management framework and involves evaluating the importance of a risk, either quantitatively, or qualitatively. Risk assessment consists of three steps, namely risk identification, risk estimation and risk evaluation. Nevertheless, the risk management framework also includes a fourth step, i.e. the need for a feedback of all the risk assessment undertakings. However, there is a lack of such feedback, which constitutes a serious deficiency in the
reduction of environmental hazards at the present time. Risk identification of local or regional hazards involves hazard quantification, event monitoring including early warning systems and statistical inference. Risk identification also involves the development of a database, where historical hazard information and their effects is included. Similarly, risk estimation involves magnitude/frequency relationships and hazard economic costs. Furthermore, risk evaluation consists of the social consequences of the derived risk and involves cost-benefit analysis and community policy. The objective of this review
paper consists of addressing meteorological hazards and extremes within the risk management framework. Analyses results and case studies over Mediterranean ecosystems with emphasis on the wider area of Greece, in the eastern Mediterranean, are presented for each of the three steps of risk assessment for several environmental hazards, such as droughts, floods, heavy convective precipitation, landslides and wildfires, using recorded datasets, model simulations and innovative methodologies. The results indicate that the risk management framework constitutes an integrated approach for
environmental planning and decision making.

**Keywords**: Environmental hazards and extremes, risk management.





## 1. Introduction

Environmental degradation is one of the major factors contributing to the **vulnerability** of environment and agriculture, because it directly magnifies the risk of natural disasters (Dalezios et al., 2017). Globally, there are several factors leading to regional vulnerability increase, such as uneven population growth, environmental degradation, increased climate variability and change, urbanization and land pressure, social inequality, technological development, or impacts of epidemics, which may also contribute to future disasters, affecting, thus, environmental sustainability (Dalezios (ed.), 2017). Moreover,

vulnerability, which could be physical, environmental, economic and social, contribute to disaster risk at regional scale. In order to ensure **sustainability** in the environmental status and agricultural production, a better understanding of the natural disasters that impact environment and agriculture is essential (Sivakumar et al., (eds), 2005). A comprehensive assessment of impacts of natural disasters on environment and agriculture requires a multidisciplinary, multi-sectoral and integral approach involving several components and factors (Dalezios et al., 2019). Priority should be given to supporting applied research,

since research is necessary to understand the physical and biological factors contributing to disasters. Community-wide awareness and capacity building programs on natural disasters, mainly for farmers and stakeholders, should also be included in any research effort. Programs for improving prediction and early warning methods, as well as dissemination of warnings, should be expanded and intensified. Moreover, efforts are required to determine the impact of disasters on natural resources. Recent research findings suggest that variability of climate, if encompassing more intense and frequent extremes, such as

major large-scale environmental hazards like droughts, heatwaves or floods, results in the occurrence of natural disasters that are beyond our socio-economic planning levels. This affects regional response capabilities, which may require new adaptation and preparedness strategies (Salinger et al. (eds.), 2005). Moreover, early warning systems of individual environmental hazards can significantly contribute to disaster prevention. Indeed, reliable forecasting schemes constitute part of integrated disaster preparedness planning for proactive risk management. Specifically, seasonal and interannual climate

forecasting contribute to mitigation and adaptation measures due to climate change. Nowadays, rapid response capabilities based on accurate early warning systems remain a priority for disaster prevention and resilience (Dalezios (ed.), 2017). In addition, technological developments increase current knowledge leading to improved risk management. Risk management means reducing the threats posed by known hazards, whereas at the same time accepting unmanageable risks and maximizing any related benefits (Smith, 2013). Indeed, risk analysis is considered an integrated methodology for

environmental hazards. Moreover, risk assessment consists of the quantitative or qualitative evaluation of risk within the risk management framework. There are three stages of risk assessment (Smith, 2013), such as risk identification, risk estimation and risk evaluation. There is also a fourth step, which is a feedback of all the risk assessment undertakings. However, there is a lack of such feedback, which currently constitutes a serious deficiency in environmental hazards reduction. Adaptation strategies to risk reduction and climate change should be based on better understanding of disasters and environmental

hazards leading to disaster risk management (Dalezios and Eslamian, 2016; IPCC, 2012). Indeed, food security and



agriculture are typical examples of sectors with links to climate, where extreme events have significant impacts (Dalezios et al., 2019).

The objective of this paper is to attempt a comprehensive presentation of the risk management framework related to environmental hazards and, specifically, to meteorological hazards and extremes. A brief presentation of the risk management framework is initially considered. This is followed by a description of the concepts of meteorological hazards. Then, environmental hazards and extremes are analysed and several case studies are presented with emphasis on the wider area of Greece, east Mediterranean.

## 2. Risk Management Framework

Figure 1 presents the main components of the risk management framework, namely risk identification, risk assessment and risk estimation, as well as risk evaluation and risk governance (from Dalezios et al., 2014), which are covered in this section. Moreover, Figure 2 presents an analytical flow chart with all the components of the risk analysis methodological procedure (Dalezios and Eslamian, 2016), which are also covered in this section.

Figure 1 (about here)

Figure 2 (about here)

### 2.1 Hazards and Disasters

**Hazard** is an inescapable part of life. Hazard is defined as "a potentially damaging physical event, *phenomenon or human activity that may cause the loss of life or injury, property damage, social and economic disruption or environmental degradation*". Hazards can include latent conditions that may represent future threats and can have different origins: natural (geological, hydrometeorological and biological) or induced by human processes (environmental degradation and technological hazards) (UN/ISDR., 2005). **Risk** has the meaning of the chance of a specific hazard to occur and is considered the product of probability and loss. As a result, risk is occasionally considered synonymous with hazard (UN/ISDR, 2005). Thus, hazard (or cause) may be defined as "*a potential threat to humans and their welfare*" and risk (or consequence) as "*the probability of a hazard occurring and creating loss*" (Smith, 2013). Unlike hazard and risk, a **disaster** is an actual happening, rather than a potential threat, thus, a disaster may be defined as "*the realization of hazard*". A more detailed disaster definition is "*an event, concentrated in time and space, in which a community experiences severe danger and disruption of its essential functions, accompanied by widespread human, material or environmental losses, which often exceed the ability of the community to cope without external assistance*" (Smith, 2013).

The term **environmental hazard** has the advantage of including a wide variety of hazard types ranging from "*natural*" (geophysical) events, through "*technological*" (man-made) events to "*social*" (human behavior) events. Specifically, it is





possible to use the following working definition of environmental hazards: "*Extreme geophysical events, biological processes and major technological accidents, characterized by concentrated releases of energy or materials which pose a large unexpected threat to human life and can cause significant damage to goods and environment*" (Smith, 2013).

**Vulnerability** is defined as "*The conditions determined by physical, social, economic and environmental factors or processes, which increase the susceptibility of a community to the impact of hazards*" (UN/ISDR, 2005). The concept of vulnerability, like risk and hazard, indicates a possible future state. Most approaches to reduce system-scale vulnerability can be viewed as expressions of either resilience or reliability. **Resilience** is defined as "*The capacity of a system, community or society potentially exposed to hazards to adapt, by resisting or changing in order to reach and maintain an acceptable level*

*of functioning and structure*". This is determined by the degree to which the social system is capable of organizing itself to increase this capacity for learning from past disasters for better future protection and to improve risk reduction measures" (UN/ISDR, 2005). **Reliability**, on the other hand, reflects the frequency with which protective devices against hazard fail (Smith, 2013).

## 2.2 Risk Identification

Risk identification involves mainly database development, risk quantification, which includes hazard modeling and monitoring, and susceptibility assessment (Dalezios et al., 2014). The potential domino effect of hazards is also analyzed, such as the links between climate and land use. A brief description of the main components follows.

**Database Development.** The database consists of historical environmental data of the study area, which are considered the input data to risk analysis (Dalezios and Eslamian, 2016). Specifically, **environmental factors** are considered**,** which

include data on meteorology, hydrology, geology and geomorphology, agronomy, topography, soil, land use/land cover, GIS and similar aspects. **Triggering factors,** which lead to hazards**,** are also considered. Moreover, the development of a **hazard inventory** is included, which is based on recorded historical disaster events. Records of the **elements at risk** for each hazard in terms of type and quantification are also included to be used in exposure analysis and vulnerability assessment (Figure 2).

**Risk Quantification.** Hazards are quantified either numerically or by modeling or even by using indicators and/or indices,

depending on the type of hazard. Moreover, the frequency of occurrence of a hazard could also be considered. Furthermore, the potential damage caused by a disaster requires the need for forecasting and monitoring in the affected region.

**Susceptibility assessment** consists of two types of analyses, namely initiation and spreading analyses (Dalezios and Eslamian, 2016). Specifically, initiation analysis consists of hazard inventory and hazard modeling based on environmental and triggering factors. Similarly, spreading analysis consists of empirical and numerical hazard modeling. Both approaches

are incorporated into risk estimation and hazard assessment (Figure 2).

## 2.3 Risk Estimation and Vulnerability Assessment

This section covers risk estimation for hazard assessment, exposure analysis and vulnerability assessment. A brief description of these components follows.





**Risk Estimation for Hazard Assessment** consists of quantification in terms of probabilistic hazard assessment and analysis

of the exposed elements at risk. Moreover, hazard severity-duration-frequency and areal extent relationships with the associated costs are considered (Dalezios et al., 2000). Several hazard scenarios with associated probabilities and indicators for magnitude, frequency and spatial extent are considered, which contributes to exposure analysis.

**Exposure analysis** consists of GIS-based spatial overlap of hazard footprints for the quantification of the elements at risk, which is incorporated into risk assessment.

**Vulnerability assessment** is based on testing, selecting and mapping indicators and uses inventories of exposed elements at risk. Uncertainty analysis of vulnerability is also considered, since several factors contribute to future scenarios. Specifically, historical damage catalogues, modeling and expert evaluation are used, as well as with indices for vulnerability assessment (Figure 2).

### 2.4 Quantitative Risk Assessment (QRA)

**Risk Analysis** contributes to QRA and consists of probabilistic scenarios, since large uncertainties exist in future risk and vulnerabilities (Figure 2). Specifically, hazard assessment involves temporal probability, severity and spatial extent (Dalezios and Eslamian, 2016). Indeed, vulnerability involves the degree of loss to each type of elements at risk and exposure refers to the spatial overlay of hazard and each element at risk. Risk analysis follows the performance of these models in terms of data requirements at different scales, as well as their effectiveness for risk assessment.

**Quantitative Risk Assessment (QRA).** QRA contributes to risk evaluation and includes probabilistic hazard assessment and the outcome of exposure and vulnerability analyses (Dalezios and Eslamian, 2016). QRA is developed through hazard and vulnerability scenarios and quantification of the elements at risk. The result is the total risk based on all the specific risks for all the hazard severities, frequencies, triggering events and elements at risk (Figure 2). The datasets are standardized and used in the risk assessment models. A Web-based platform could be used, where all the components are integrated into the

risk management framework. The qualitative risk assessment is used, which involves indices, in case that risk cannot be quatified (Figure 2).

### 2.5 Risk Evaluation

**Risk Evaluation.** Risk evaluation refers to the loss related to each disaster event leading to risk reduction (Dalezios and Eslamian, 2016). All the risk management options and optimal tools are analyzed based on the previous risk scenarios. Risk

evaluation involves cost-benefit analysis and policy issues. This part also involves methods or indicators for the estimation of adaptation options at different spatial and temporal scales. Indeed, risk evaluation involves hazard and risk information to be integrated into Environmental Impact Assessment (EIA) and Strategic Environmental Assessment (SEA), land use planning, cost-benefit analysis of adaptation options for the development of mitigation measures, early warning and emergency preparedness plans (Figure 2).



**Development of Decision Support System (DSS).** DSS is considered a very useful web-based multi-scale and interactive tool. Indeed, risk evaluation performance is incorporated into the DSS. The development of the DSS follows several phases starting from the intelligence phase, which is followed by the design phase and then by the decision phase (Dalezios and Eslamian, 2016). Specifically, the decision phase involves cost benefit analysis (CBA), physical planning approaches, social impact assessment, environmental impact assessment (EIA) and spatial multi-criteria evaluation (MCE), which includes

hazard and risk information for spatial planning (Figure 2).

### 2.6 Risk Governance

**Feedback of risk reduction.** Successful risk governance leads to the effectiveness of the risk reduction measures. Indeed, risk governance attempts to integrate all the rules, mechanisms and processes, which are implemented within the risk management framework. Reliable risk governance strategies are based on risk evaluation, which relies on both good science

and good judgment (Dalezios (ed.), 2017; Dalezios and Eslamian, 2026). Both the QRA and the relevant aspects of risk perception should be considered. The methods of hazard and risk assessment should be demonstrated to local stakeholders/end-users, where the target is to achieve an agreement on risk reduction measures. However, there is currently lack of feedback, which signifies a serious deficiency in the reduction of environmental hazards (Smith, 2013).

**Dissemination of results and public awareness.** The feedback could justify the level of public awareness and response by

the Authorities. Suitable dissemination information and training could be developed for different stakeholders. Dissemination tools and activities can be employed for improving the level of public awareness and the extent of information spreading.

### 3. Meteorological and Environmental Hazards: Case studies

Each hazard is somehow considered unique. For instance, droughts show a slow escalation, however, populations and large

regions could be affected for extended periods of time (Dalezios et al., 2017). On the other hand, flash floods or tornadoes are short-lived, violent events, affecting relatively small areas. Moreover, extreme meteorological events can result into multiple hazards. For example, a tropical storm, besides heavy rain and high winds, can lead to flooding. Furthermore, in temperate latitudes, severe summer extreme events, such as thunder and lightning storms or tornadoes, can be accompanied by heavy hail and flash floods. In addition, avalanches occurring on some mountain slopes and high runoff or flooding

during the melt season can be caused by winter storms with high winds and heavy snow or freezing rain.

According to the World Meteorological Organization (WMO, 2006), several environmental hazards originate from meteorological events, such as tropical and extra-tropical cyclones, tornadoes, thunderstorms, lightning, hailstorms, high winds, snowstorms, freezing rain, dense fog, thermal extremes and droughts (Nastos and Dalezios, 2016). Similarly, there are several other environmental hazards, which are associated with weather and water, such as floods and flash floods, storm





surges, high waves at sea, sand storms, wildfires, smoke and haze, landslides and mudslides, avalanches and desert locust swarms (Nastos and Dalezios, 2016).

Advances in environmental rely on recorded data sets, model simulations and innovative methodologies. In the following, meteorological and environmental hazards and extremes are presented, by means of their definitions and related citations. Furthermore, the analysis is enriched with characteristic case studies, mainly over the wider area of Greece.

**3.1 Heat waves**

The conventional definition of a heat wave is referred to the combination of an abnormal and uncomfortable period of hot weather with high air and humidity, with a typical duration of about two days (Koppe et al., 2004). It is mentioned that until these days the World Meteorological Organization has not yet announced a specific definition of heat waves. Heat wave is certainly a meteorological phenomenon, however, there is always assessment of the related impacts on humans. As a result,

it is sometimes better to consider the human sensation of heat against determining specific thresholds of meteorological parameters. There are different considerations and views. For instance, a heat wave can be considered as an extended period of unusually high atmosphere-related heat stress, which temporary affects lifestyle habits and causing health related problems to humans (Robinson, 2001). Moreover, continuous global warming is very likely to cause heat waves to occur with a higher frequency and duration towards the end of 21th century (Beniston et al. 2007; IPCC 2013; Tolika, et al., 2014).

It has been proven that 'Mega-heat waves' surpassed the 500-yr long seasonal temperature data over approximately 50% of Europe (Barriopedro et al., 2011; Katsafados et al., 2014), like the cases of 2003 and 2010, respectively. During the event of summer 2003, it has been noted the excess of the temperatures of 1961-1990 mean by about 3ºC in a large area of central Europe and this fact corresponds to an enormity of up to 5 standard deviations (Schär et al., 2004). For more than a week in a great number of European cities the daily maximum temperature exceeded 35 ºC. This fact caused about 70,000 excess

deaths in southern, western and central regions of Europe (Robine et al., 2006; Vandentorren et al., 2006). A particular effort for the identification of heat waves in Athens, Greece, has been conducted by Matzarakis and Nastos (2011), with the use of the Physiologically Equivalent Temperature (PET), a human thermal index based on the energy balance of the human body. For the scope of quantifying the duration of heat waves and their impacts (Figure 3), they used consecutive days (three and more) and the record of duration of each heat wave, for PET $\geq$ 35 °C, which is the threshold of extreme heat stress

(Matzarakis et al., 1999) and $T_{amin} \geq 23$ °C, which represents the threshold of PET values of thermal neutrality (Nastos and Matzarakis 2008).

Figure 3 (about here)




It is assessed that for the number of consecutive days with respect to PET ≥ 35 °C, there has not been evidence of a clear pattern, though since 1983 there has been observed a statistically significant increasing trend of the maximum duration of heat waves within the year (b=1.33 days/year, p=0.000). Indeed, since 1983 the annual number of heat waves (HW) evolves a statistically significant trend (b=0.26 HW/year, p=0.000). As far as the consecutive days with $T_{amin} ≥ 23$ °C are concerned,

a statistically significant trend for the number of heat waves within the year (b=0.15 HW/year, p=0.048) appears since 1983, while for the maximum duration of heat waves within the year (b=0.07 days/year, p=0.344), a statistically significant trend cannot be mentioned.

## 3.2 Extreme air temperature indices

There are three main categories for the division of extreme air temperature indices: absolute, percentile and duration indices,

defined by the joint CCl/CLIVAR/JCOMM Expert Team (ET) on Climate Change Detection and Indices (Alexander et al., 2006; Burić et al., 2014). The concern of the absolute indices has as follows: summer days, SU25 (number of days with daily maximum temperature above 25 °C); tropical days, SU30 (number of days with daily maximum temperature above 30 °C); tropical nights, TR20 (number of days with daily minimum temperature above 20 °C); frost days, FD0 (days with absolute minimum temperature below 0 °C); maximum of daily maximum temperature, TXx (Let $Tx_j$ be the daily maximum

temperatures in period $j$. The maximum of daily maximum temperature is then $TXx_j = max(Tx_j)$; maximum of daily minimum temperature, TNx (Let $Tn_j$ be the daily minimum temperatures in period $j$. The maximum of daily minimum temperature is then $TNx_j = max(Tn_j)$. The percentile indices regard: warm days, TX90p (the number of days with daily maximum temperature above the 90[th] percentile calculated for each calendar day, on basis of 1961-1990, using 5 day running window); warm nights, TN90p (the number of days with daily minimum temperature above the 90[th] percentile calculated for each

calendar day, on basis of 1961-1990, using 5 day running window); cold days TX10p (the number of days with daily maximum temperature below the 10[th] percentile calculated for each calendar day, on basis of 1961-1990, using 5 day running window); cold nights TN10p (the number of days with daily minimum temperature below the 10[th] percentile calculated for each calendar day, on basis of 1961-1990, using 5 day running window). The third category of extreme air temperature referes the duration index WSDI (Let $Tx_{ij}$ be the daily maximum temperature on day $i$ in period $j$ and let $Tx_{in90}$

be the calendar day 90[th] percentile centred on a 5-day window; Zhang et al., 2005). Thus, the number of days per period is calculated where, in intervals of at least 6 consecutive days: $Tx_{ij} > Tx_{in90}$. It is mentioned that a spell can continue into the next year for spell/duration indices and is counted against the year in which the spell ends. The examination of future projected patterns for extreme indices of air temperature in Greece has been conducted by Nastos and Kapsomenakis (2015), through the use of simulations from six RCMs employed for the reference (1961-1990), near (2031-2050) and far (2071-

2100) future periods. They concluded that temperature extremes presented widespread significant changes associated with projected warming, in particular in the far future under SRES A1B. A remarkable contrast of land-maritime air temperature extremes was the result of the simulations, as land and sea are characterized by different thermal characteristics. It is depicted in Figure 4, the ensemble means of Maximum Daily Maximum Temperature (TXx) (left graphs) and Maximum



Daily Minimum Temperature (TNx) (right graphs) for the reference period (A, B), along with changes of near future (C, D)
and far future (E, F) from the reference period. To be more specific, the increase of TXx in the near future is projected to be
2.4 ℃ – 3.0 ℃ over land against 2.0 ℃ – 2.2 ℃ over sea (Figure 4C), albeit higher increase is expected in the far future;
namely 4.4 ℃ – 5.4 ℃ for continental Greece and 3.6 ℃ – 4.2 ℃ over sea (Figure 4E). As for the increase of TNx in the
near future, there is a projection of 2.6 ℃ – 2.8 ℃ over land against 2.0 ℃ – 2.4 ℃ over sea (Figure 4D), while for the
projections in the far future higher increase is expected; namely 4.6 ℃ – 5.4 ℃ for continental Greece and 3.6 ℃ – 4.4 ℃
over sea (Figure 4F).

Figure 4 (about here)

## 3.3 Tornadoes and waterspouts

The tornado is a violently whirling column of air in contact with the ground or hanging from a cumulonimbus and often (but
not always) visible as a funnel cloud. The tornadoes and waterspouts are identical phenomena, the first definition is used
over land and the second over sea. The horizontal extent of the tornado reaches even 250 m, and the speed of movement is
relatively small (8-20 m s$^{-1}$). The speed of the spinning column of air in the central region reaches 100 m s$^{-1}$, but can also
exceed these speeds reaching 200 m s$^{-1}$. At the same time the vertical movements of the air are very powerful. The pressure
gradient from the periphery to the center of tornado presents remarkable fall and can reach 25 hPa, having as a result to
intensify the rotational movement of the wind. The path traveled by a tornado is relatively small, 10 km, reaching in specific
cases 200 km, having a life period of 4-5 hours. The passage of a tornado causes major damage due to stormy winds and the
sharp drop in atmospheric pressure.

Tornadoes are extreme phenomena associated with severe convective storms (Nastos and Dalezios, 2016). The Greek
philosopher Aristotle (384-322 BC) in "Meteorologica" presented perhaps the most renowned exposition of natural extreme
phenomena: "*So the whirlwind originates in the failure of an incipient cyclone to escape from its cloud. It is due to the
resistance the eddy generates and emerges when the spiral descends to the earth dragging along the cloud that cannot shake
off. When blowing in a straight line it carries along whatever comes by in a circular motion and overturns and snatches up
whatever it meets*" (Lee, 1952). Tornadoes occur in many parts of the world (Fujita, 1973). Several publications during the
last several decades have presented historical records concerning tornadic activity (e.g., Meaden, 1976; Tomming et al.,
1995; Peterson, 1998; Reynolds, 1999; Tyrrel, 2003; Macrinoniene, 2003; Dotzek, 2003; Nastos and Matsangouras, 2010;
Gayà et al., 2000; Brázdil et al., 2012; Rahuala et al., 2012; Haghroosta, et al., 2014). As far as tornadic activity over Greece
is concerned, a comprehensive spatial distribution of a total of 612 events (171 tornadoes, 374 waterspouts and 67 funnel
clouds), recorded on 405 days was presented (Matsangouras et al., 2014a;2014b: Nastos & Matsangouras, 2014), as there
were several days with multiplied events, within the period 1709-2012 (Figure 5). This study give evidence that even in an



eastern Mediterranean region these fury atmospheric phenomena are abundant, causing catastrophic impacts on
infrastructures and in many cases loss of life.

Figure 5 (about here)

Golden (1974a; 1974b; 1977; 2003) provided the description of fundamental processes of waterspout formation, as well as
the identification of the water-surface signatures of waterspouts regarding their development stage and intensity. Five stages
of waterspout was analysed: (1) the dark spot, (2) the spiral pattern, (3) the spray ring, (4) the mature waterspout and (5) the
decay stage. It is determined that there is either cyclonically or anticyclonically rotation of waterspouts, with a range of
surface diameters between 5 and 75 m; local horizontal wind shear provokes their vorticity. It has been observed that air
temperature and pressure perturbations within waterspouts vary from 0.2 to 2.5 K and from 10 to 90 hPa, respectively. The
common formation of waterspouts mostly takes place under convective clouds, while waterspout genesis reveals preferable
tendency in regions of local horizontal shear lines, separating the updrafts from downdrafts (Golden, 1974a; Leverson et al.,
1977; Hess and Spillane, 1990), but it has to be accentuated that this condition is a necessary but not sufficient one for the
formation of waterspout (Simpson et al., 1986).


## 3.4 Medicanes

Medicanes are Mediterranean Tropical Like Cyclones (TLC), which are meso-scale extreme low-pressure systems,
resembling the structure of tropical cyclones, as they captured by satellites (Nastos and Dalezios, 2016). Two areas have
experienced larger number of medicane formation, the western Mediterranean and the wider area of the Ionian Sea with
limited occurrence in Aegean Sea and eastern Mediterranean. Figure 6 illustrates the seasonal geographical distribution of
medicane occurrence over Mediterranean within the period 1969-2014 (Nastos et al., 2018).

Figure 6 (about here)


Their intensity appears much weaker than tropical hurricanes; however, some of them have reached tropical hurricane
strengths (Akhtar et al., 2014). Similarly, their genesis is triggered when an upper-level cut-off low is advected over an area,
which results in air mass lifting and cooling causing convective instability (Emanuel, 2005). Moreover, their structure and
evolution are considered very significant (Pytharoulis et al., 2000; Homar et al., 2003; Moscatello et al., 2008), as well as the
model physics in simulating the structure and intensity (Miglietta et al., 2015). Indeed, these meso-scale systems with
diameter usually less than 300 km have a rounded structure and a warm core, as well as intense low sea level pressure



(Businger and Reed, 1989). Moreover, strong winds, heavy precipitation and thunderstorms are associated with the incidence of medicanes, causing occasional severe damages in private property, agriculture and communication networks, or resulting in flooding of populated areas, posing a risk to human life (Figure 7).


Figure 7 (about here)

In a recent study (Mylonas et al. 2019), the higher spatial horizontal resolution of a TLC event, south of Sicily on November 7–8, 2014, through a physics parameterization sensitivity analysis, allows for improved simulations in most setups that were 325 tested in terms of trajectory and TLC structure.

## 3.5 Extreme precipitation indices and heavy convective precipitation

**Extreme precipitation indices**. The extreme precipitation indices can also be classified into three classes: absolute, percentile, and duration indices, as defined by the joint CCl/CLIVAR/JCOMM Expert Team (ET) on Climate Change 330 Detection and Indices (Alexander et al., 2006). The absolute threshold indices concern: number of heavy precipitation days (number of days with daily precipitation amount above 10mm), number of very heavy precipitation days (number of days with daily precipitation amount above 20mm) and simple daily intensity index (daily precipitation amount on wet days in a period per number of wet days in the period) (Benhamrouche et al., 2015). The percentile indices concern: very wet days (the number of days with daily precipitation amount above the 95$^{th}$ percentile from the examined period) and extremely wet 335 days (the number of days with daily precipitation amount above the 99$^{th}$ percentile from the examined period). The duration indices concern consecutive dry days (the largest number of consecutive days with daily precipitation amount below 1 mm) and consecutive wet days (the largest number of consecutive days with daily precipitation amount above 1 mm). In a recent study, temporal trends and spatial patterns in precipitation and temperature and their extremes in the eastern Mediterranean and Middle East region (EMME) have been presented, using output from the Hadley Centre PRECIS climate model 340 (Kostopoulou et al., 2014). The model projects drying trends by 5–30 % in annual precipitation towards the end of the 21st century, with the number of wet days decreasing at the rate of 10–30 days year$^{-1}$, while heavy precipitation is likely to decrease in the high-elevation areas by 15 days year$^{-1}$.

**Heavy convective precipitation**. Heavy precipitation typically occurs with moist deep convection. Initially, a cloud is formed through condensation of the excess water vapor in rising air parcels. Then, the heat released through this 345 condensation can help to sustain the convection by warming the air further and making it rise still higher, causing more water vapor to condense, so the process feeds on itself. There are three required ingredients in order to produce moist deep convection: the environmental lapse rate must be conditionally unstable; there must be enough lifting so that a parcel will reach its level of free convection; and there must be enough moisture present that a rising parcel's associated moist adiabat has a level of free convection (Doswell et al., 1996). In mid-latitudes, convective precipitation is associated with cold fronts,



squall lines, and warm fronts in very moist air. Similarly, graupel and hail indicate convection. On the other hand, in the
       tropics, precipitation produced solely through condensation and accretion of liquid, is considered important (Rogers 1967,
       Houze 1977). However, the warm rain process may also play a critical role in heavy convective precipitation events in
       midlatitudes as well, resulting in many flash floods and landslides (Segoni, et al., 2014a: 2014b). Several researchers have
       noted the importance of convection and especially, mesoscale convective systems in producing warm season precipitation.

Heideman and Fritsch (1984) estimated that about half of warm season precipitation over the United States is produced
       directly by mesoscale systems or phenomena. In long lasting, circular shaped, convective systems, using satellite imagery
       found that such systems accounted for approximately 30 to 70% of warm season precipitation between the Rocky Mountains
       and Mississippi River. Extreme weather events, including heavy rain, lightnings, waterspouts and severe wind gusts occur
       due to the interaction between large-scale environmental conditions and local conditions, related to pure convection.

Specifically, Nastos et al. (2014) showed that the seasonal distribution of Cloud-to-Ground (CG) lightning activity frequency
       coincide well with the regional climatic convective characteristics of Greece; namely CG strokes are dominant over land and
       coastal areas during summer and spring, against over warm water bodies of Aegean and Ionian Seas, during the other
       seasons. Convective conditions and summer showers are a frequent phenomenon in the greater Athens area, Greece (Feloni
       et al., 2019). A recent research campaign concerns the COnvective Precipitation Experiment (COPE), which was a joint UK-

US field campaign held during the summer of 2013 in the southwest peninsula of England, designed to study convective
       clouds that produce heavy rain leading to flash floods. The clouds form along convergence lines that develop regularly due
       to the topography. The overarching goal of COPE is to improve quantitative convective precipitation forecasting by
       understanding the interactions of the cloud microphysics and dynamics and thereby to improve NWP model skill for
       forecasts of flash floods (Leon et al., 2016). Besides, WRF simulations were carried out to examine the sensitivity of the

rainfall distribution in and around the urban area to different urban land surface model representations and urban land-use
       scenarios (Alexakis et al., 2014). Simulation results suggest that urbanization plays an important role in precipitation
       distribution, even in settings characterized by strong large-scale forcing (Yang et al., 2014). Nastos et al. (2017) concluded
       that the urbanization of Athens, Greece, due to the rapid increase of the population and the number of vehicles the last
       decades, had remarkable impacts on the mean annual rain intensity and annual number of days for rain events over 10mm,

20mm and 30mm. The analysis of the rain intensity for Athens (Figure 8, left graphs), revealed a statistically significant
       (C.L. 95%) positive trend (+0.03mm/h/year) for rain events over 10mm, during the examined period 1930-2004, while
       stronger trends, statistically significant (C.L. 95%) within the period 1990-2004, with respect to the rain threshold of 10mm
       (+0.46mm/h/year) and 20mm (0.48mm/h/year) appear. Similar results have been found with respect to the annual number of
       days with daily rain totals ≥ 10mm, 20mm and 30mm (Figure 8, right graphs). Many studies have given evidence that the

Urban Heat Island (UHI) triggers convective precipitation in Atlanta (Bornstein and Lin 2000), in Beijing City (Guo et al.
       2006), in Tokyo (Yonetani 1982), in London, in Ankara (Cicek and Turkoglu 2005), in Athens (Nastos and Zerefos, 2007;
       Giannaros et al., 2014).





Figure 8 (about here)

### 3.6 Droughts

Drought is a natural, casual and temporary state of continuous decline in precipitation and water availability in relation to normal values, spanning a considerable period and covers a wide area. It is discriminated into meteorological, hydrological and agricultural drought (Dalezios et al., 2019; Dalezios, 2018). Drought is a regional phenomenon, which is characterized by its severity, duration and areal extent (Tsakiris et al., 2007). Several sectors of the economy, environment and society are affected by drought (Wang, 2005; Mechler et al., 2010; Dalezios et al., 2012). Historically, the identification of dry areas has been considered two millennia ago (Nastos and Dalezios, 2016). Specifically, the classical Greek thought acknowledged that the latitude affects the arid, temperate and cold zones of the earth. There was a perception that the arid climates in small latitudes were dry (Nastos et al., 2013). The evaluation of drought is accomplished by drought indices, the most important of which and widely used are the Aridity Index (AI) (UNESCO, 1979), the  Standardized Precipitation Index (SPI) (McKee et al. 1993), Palmer Drought Severity Index (PDSI) (Palmer 1965) and Reclamation Drought Index (RDI) (Weghorst 1996). The development of Earth observation satellites from the 1980s onwards promoted drought monitoring and detection. The most prominent vegetation index is certainly the Normalized Difference Vegetation Index (NDVI; Tucker, 1979) that was first applied to drought monitoring by Tucker and Choudhury (1987). The index NDVI, by itself, does not depict drought or not drought conditions, but the severity of drought can be defined as the deviation from the mean NDVI value of a long period (DEVNDVI). Nastos et al. (2013) studied the spatiotemporal patterns of the Aridity Index (AI) in Greece, per decade, during 1951-2000 and the projected changes in ensemble mean AI between the period 1961-1990 (reference period) and the near (2021-2050) and far future (2071-2100), simulated by a number of Regional Climatic Models (RCMs), within the ENSEMBLE European Project under SRES A1B. They illustrated a progressive shift from the "humid" class, which characterized the wider area of Greece, towards the "sub-humid" and  "semi-arid" classes appeared in the eastern Crete Island, the Cyclades complex, the Evia and Attica, that is mainly the eastern Greece, most pronounced within the period 1991-2000 (Figure 9). Drier conditions are anticipated to appear in subregions of Greece (Attica, eastern continental Greece, Cyclades, Dodecanese, eastern Crete island and northern Aegean).

Figure 9 (about here)

Similar results have been extracted by Polychroni and Nastos (2017), who found decreasing trends of the annual SPI in Greece and western Turkey, against increasing trends in north-eastern Europe and north-western Africa, both statistically significant (at 95% C.L.), during the period 1981-2010 (Figure 10). The atmospheric circulation, by means of North Atlantic Oscillation Index (NAOI) and North Sea Caspian Pattern Index (NCPI), seems to influence SPI variability, making the climate drier or wetter depending on the phase of the indices.





Figure 10 (about here)

Dalezios et al. (2014) identified the agricultural drought in Thessaly, which is the major agricultural drought-prone region of
Greece, characterized by vulnerable agriculture, by the implementation of the vegetation health index (VHI), which is based
on satellite data of temperature and the normalized difference vegetation index (NDVI). The results show that agricultural
drought appears every year during the warm season in the region. The severity of drought is increasing from mild to extreme
throughout the warm season, with peaks appearing in the summer. Similarly, the areal extent of drought is also increasing
during the warm season, whereas the number of extreme drought pixels is much less than those of mild to moderate drought
throughout the warm season.

## 3.7 Wildfires

There is a steady increase in the frequency of large wildfires and the total area burned, mainly due to global warming (Nastos
and Dalezios, 2016). It is expected that drier conditions will increase the probability of fire occurrence, although more than
four out of every five wildfires are caused by people. Furthermore, more fuel is expected to become available for forest fires,
since warmer and drier conditions are conducive to widespread beetle and other insect infestations, resulting in broad ranges
of dead and highly combustible trees (Joyce et al., 2008). Moreover, longer fire seasons are expected, since spring runoff is
expected to occur earlier, summer heat will build up more quickly, and warm conditions will extend further into fall
(Running, 2006). In addition, increased frequency of lightning is expected, since thunderstorms become more severe (Price,
2009). Similarly, droughts, heat waves and cyclical climate changes, such as El Niño, can also have a dramatic effect on
wildfires risk. There are several reliable fire danger rating systems used worldwide, including the Canadian Forest Fire
Weather Index System (CFFWIS) used in Canada (van Wagner, 1987), the National Fire Danger Rating System (NFDRS)
used in the USA (Deeming et al., 1977) and the McArthur Forest Fire Danger Index (FFDI) used in Australian forests (Mc
Arthur, 1967). In Europe, some well-known indices include the Finnish Fire Index (FFI), developed by the Finnish
Meteorological Institute (Venäläinen and Heikinheimo, 2003); the Portuguese index (ICONA, 1988); and the Italian index
(IREPI) proposed by Bovio et al. (1984). Karali et al. (2014) evaluated the Canadian Fire Weather Index (FWI) over Greece,
by suggesting three critical fire risk threshold values: FWI = 15 for western Greece, FWI = 30 for northern Greece and FWI
= 45 for eastern Greece. Future fire risk projections suggest a general increase in fire risk over the domain of interest, with a
very strong impact in the eastern Peloponnese, Attica, central Macedonia, Thessaly and Crete. It is expected that 15 to 20
critical fire risk days will be added in western and northern Greece. For eastern and southern Greece, the increase reaches up
to 10 days per year. For the distant future, the same pattern applies, with an increase of 30 to 40 days for western and
northern Greece and 20 to 30 for eastern and southern Greece (Figure 11).

Figure 11 (about here)



## 4. Summary and Conclusions

The objective of this review paper is twofold: to present the risk management framework of meteorological hazards and extremes, and to analyze the results and case studies for each of the three steps of risk assessment for several meteorological and environmental hazards in the wider area of Greece, east Mediterranean. More specifically, a comprehensive presentation of the risk management framework related to meteorological hazards and extremes is introduced followed by a description of the concepts of meteorological hazards. On the other hand, the analysis is enriched with characteristic case studies, mainly

over the wider area of Greece. The readers of this paper will benefit to understand the physical systems and environmental processes in an integrated manner. Last but not least, the authors consider that this scientific effort contributes to the existed knowledge of modeling and assessing environmental hazards and extreme events, appeared mainly in the wider area of Greece in the eastern Mediterranean, a vulnerable area, taking into consideration the impacts of climate change on the intensity and frequency of large-scale environmental hazards, such as floods, droughts, or heatwaves.

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









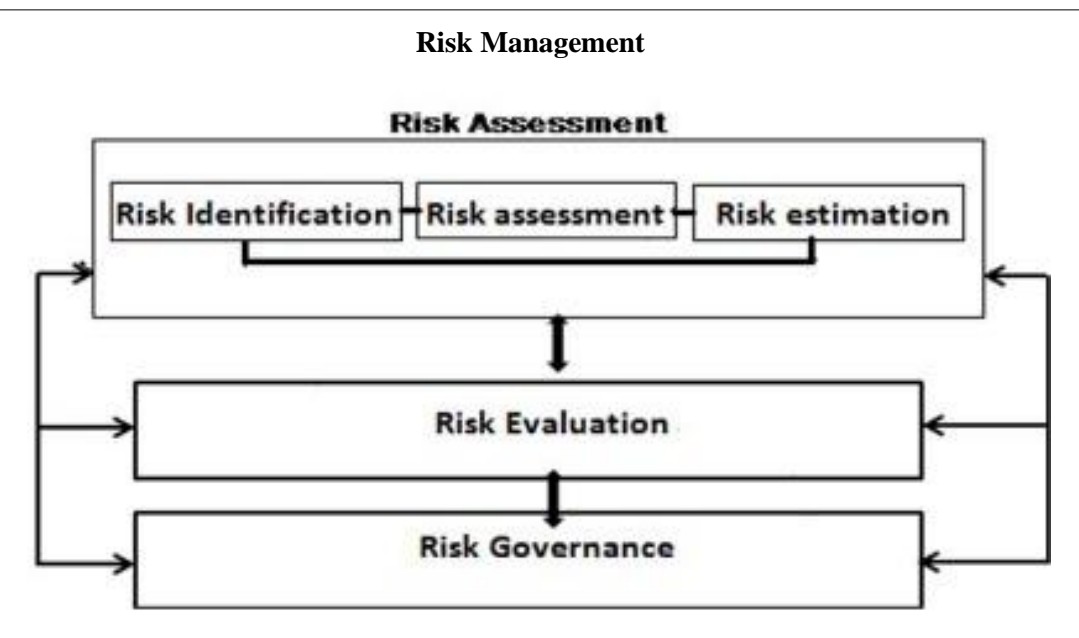

**Figure 1. Components of Drought Risk Management (Adapted from Dalezios et al., 2014.**





**A : RISK IDENTIFICATION**

**A1. DATABASE**

| A1.1 Environmental Factors | | A1.2 Triggering Factors | A1.3 Hazard Inventory | A1.4 Elements at risk |
|---|---|---|---|---|
| Geology Geomorphology Soils Topography GIS | Land Cover / Meteorology / Hydrology | Different for each hazard e.g. Rainfall | Landslides Floods Droughts Forest Fires Hail etc | Different for each hazard, e.g. for floods: buildings, roads etc. |

**A2. SUSCEPTIBILITY ASSESSMENT**

**A2.1 Initiation Analysis**
Inventory, heuristic, statistical, physical modeling

**A.2.2 Spreading Analysis**
Empirical, Analytical, Numerical models

**B : RISK ESTIMATION**

**B1 : HAZARD ASSESSMENT**

Spatial Probability

Temporal Probability

Intensity Probability

Magnitude – Frequency Analysis

Hazard assessment:
$P_s$, $P_t$, and $P_m$ for different return periods of trigging events and hazard types

**B2 : EXPOSURE ANALYSIS**

GIS Analysis
- Spatial overlay of hazard footprints
- Elements at risk

**B3 : VULNERABILITY ASSESSMENT**
Magnitude - loss relationships ( heuristic, empirical, analytical)
Vulnerability Curve, Vulnerability Matrix, Indicators, V=1

**C : RISK ASSESSMENT**

**C1 : Specific Risk Scenarios**

| Probability of hazard scenarios | | | | |
|---|---|---|---|---|
| Scenarios | $P_T$ | Loss (V*A) | | |
| | | Min | Average | Max |
| 1 | 0.1 | | | |
| 2 | 0.02 | | | |
| 3 | 0.01 | | | |
| etc | etc | | | |

$\Sigma$  Vulnerability of elements at risk  x  Quantification of amount of exposed Elements at risk
- $\Sigma$  All hazard types
- $\Sigma$  All hazard intensities
- $\Sigma$  All return periods
- $\Sigma$  All triggering events
- $\Sigma$  All elements at risk

**C2 : Quantitative Risk Assessment**
Combining Specific Risk Curves

Total Risk (Probability / Loss) — Max / Min

F-N Curves (Probability / Number (N) of Fatalities)

**C3 : Qualitative Risk Assessment**
If risk cannot be quantified

Hazard Index    Vulnerability Index
Risk Index

Spatial Multi-Criteria Evaluation (SMCE)

**D : RISK EVALUATION**

| Cost-Benefit Analysis CBA | Early Warning Systems | Spatial Planning |
| Mitigation Measures | Preparedness Planning | Environmental Impact Assessment |
| Decision Support System | | Strategic Impact Assessment |

**E : RISK GOVERNANCE**

| E1 : FEEDBACK OF RISK REDUCTION | E2 : PUBLIC AWARENESS |

**Figure 2. Flow chart of Risk Analysis Methodological Procedure (Adapted from Dalezios and Eslamian, 2016)**




**Figure 3. Consecutive number of days with PET ≥ 35 °C (upper graph) and Tamin ≥ 23 °C (lower graph) for Hellenikon/Athens, during the period 1955-2001. The number in the bars indicates the duration in days for each heat wave recorded (Adapted from Matzarakis and and Nastos, 2011)**


(A) TXx for the period 1961-1990

(B)TNx for the period 1961-1990

(C) TXx change 2031-2050 minus 1961-1990

(D) TNx change 2031-2050 minus 1961-1990

(E) TXx change 2071-2100 minus 1961-1990

(F) TNx change 2071-2100 minus 1961-1990

**Figure 4. Spatial distribution of ensemble means of Maximum Daily Maximum Temperature (TXx) (left graphs) and Maximum Daily Minimum Temperature (TNx) (right graphs) for the reference period (A, B), along with changes of near future (C, D) and far future (E, F) from the reference period. The color scale concerns Celsius degrees (ºC). (Adapted from Nastos and Kaposmenakis, 2015)**





**Figure 5. Spatial variability of tornadoes, waterspouts and funnel clouds over Greece for the period 1709-2012. (Numerous of tornadic events lie under the tornadic type symbols, due to low resolution of the image) (Adapted from Matsangouras et al. 2014a)**



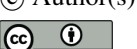


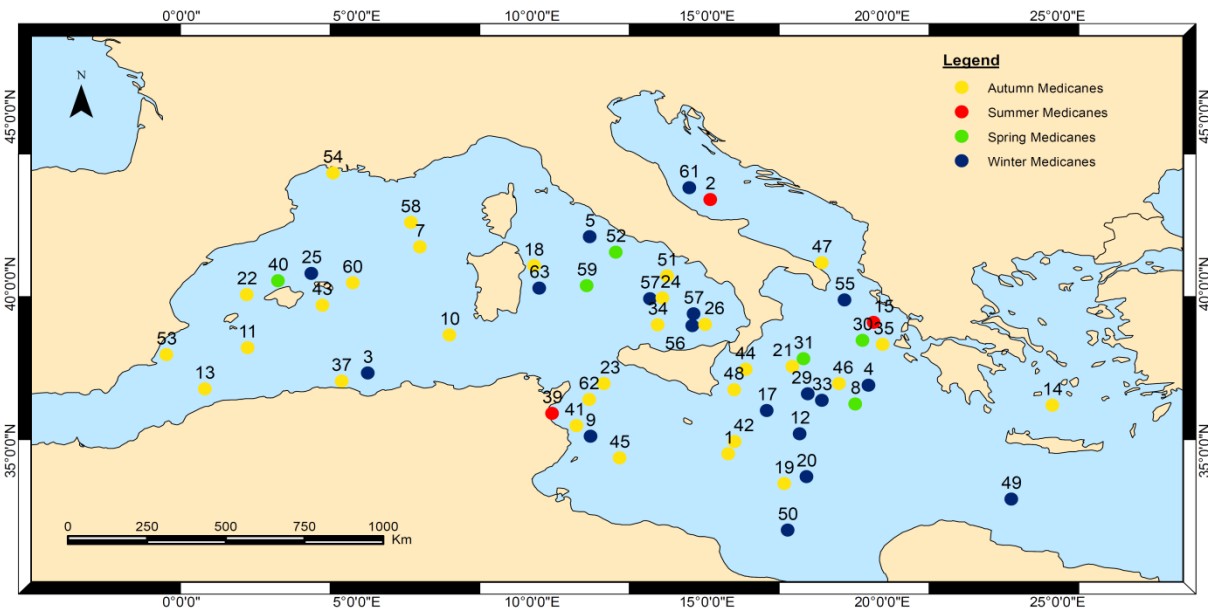

**Figure 6. Seasonal geographical distribution of medicane occurrence (yellow color for autumn, red color for summer, green color for spring and blue color for winter) over Mediterranean during the study period 1969-2014 (Adapted from Nastos et al., 2018)**





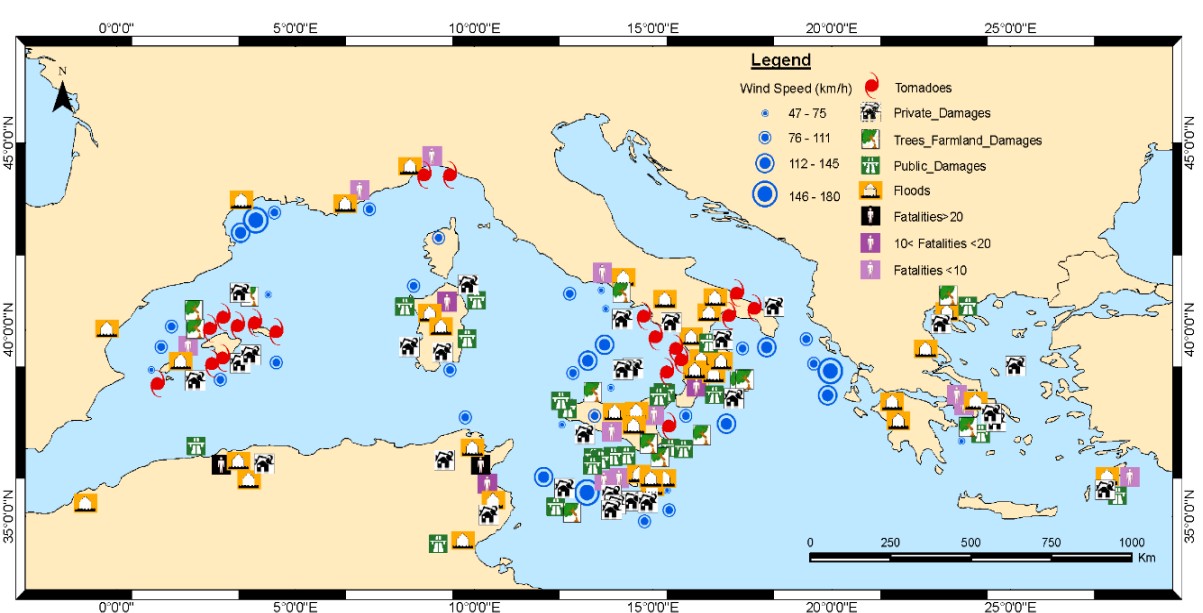

**Figure 7. Geographical distribution of medicane impacts over Mediterranean during the study period 1969-2014 (Adapted from Nastos et al., 2018)**



**Figure 8. Mean annual rain intensity (mm/h) (left graphs) and annual number of days (right graphs), with daily rain totals ≥ 10mm, 20mm and 30mm, along with the 9 points moving average fitting (red line), for Athens, during the period 1930-2004 (Adapted from Nastos et al., 2017)**




**Figure 9. Spatial distribution of the Aridity Index per decade for the period 1951-2000, based on stations' data (Adapted from Nastos et al., 2013)**

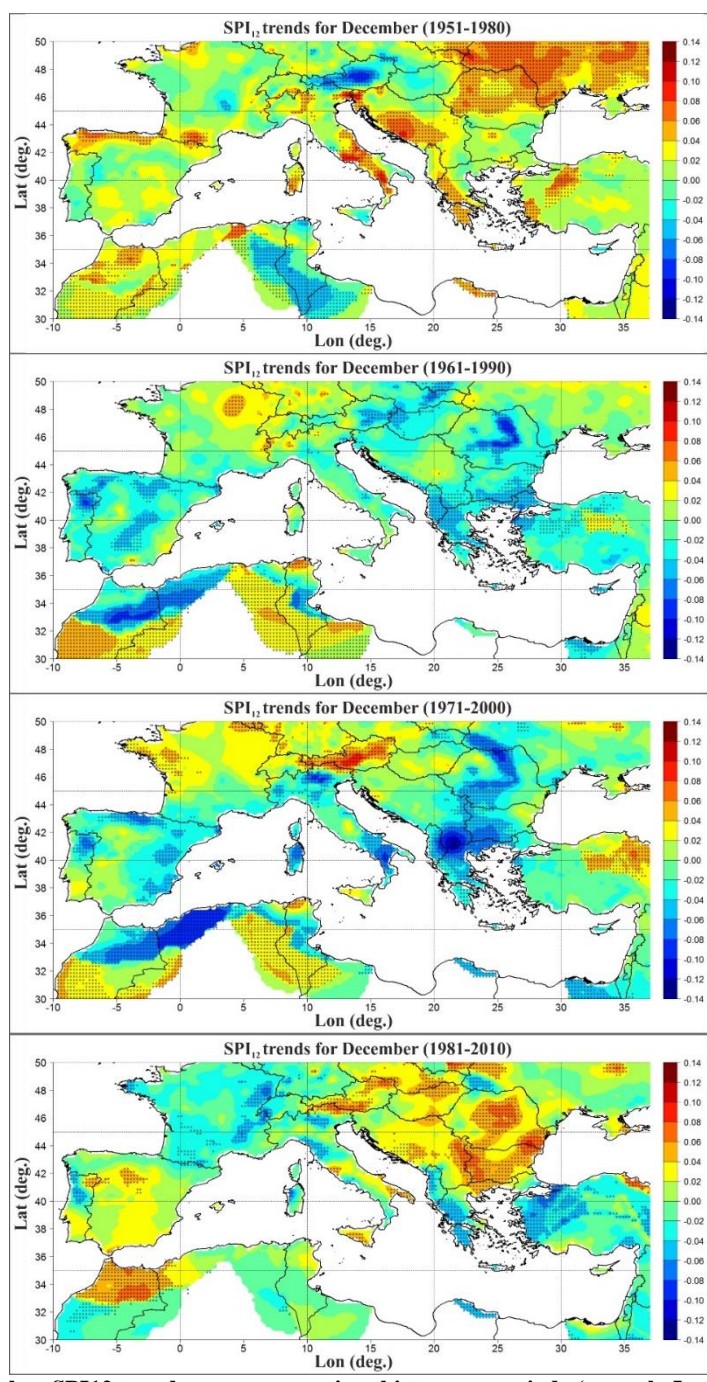

**Figure 10. Evolution of December SPI12 trends over consecutive thirty-year periods (annual; January to December). The values with asterisk (\*) refer to statistically significant trends at 95% cl. (Adapted from Polychroni and Nastos, 2017)**



**Figure 11. Mean number of critical fire risk days for the control period (1961–1990), (a, b, c), differences between the near future**
815 **(2021– 2050) and the control period, (d, e, f) and differences between the distant future (2071–2100) and the control period (g, h, i). Columns correspond to the mean number of days with FWI values above the critical fire risk threshold for different subregions: western Greece (a, d, g), northern Greece (b, e, h), and eastern/southern Greece (c, f, i) (Adapted from Karali et al., 2014)**