# Peer review of "RISK MANAGEMENT FRAMEWORK OF ENVIRONMENTAL HAZARDS AND EXTREMES IN MEDITERRANEAN ECOSYSTEMS"

_Natural Hazards and Earth System Sciences, 2020_

## Referee Comment (RC1) · Anonymous Referee #1 · 27 Sep 2020

The paper presents a double review: on one side there is the presentation of a framework for risk management, while on the other there is a review of extreme meteorological hazards occurring mainly (but not only) in the wider are of Greece and Eastern Mediterranean.

As for the first review, it would be important to control used terminology and show similarities/dissimilarities with most recent literature. Further it would be important to describe the entire workflow in a more fluid way and less as a list of definitions.

As for the second review, it presents several recent findings for the considered hazard and and case study area. However, this part is not connected with the previous one.

It would be important to connect the two parts to make understand why and how they are connected.

Below some specific comments to the text and to the figures.

- Page 2 line 35: I would not start the paper with that sentence, rather I would mention the several factors at the global scale and then say that environmental degradation is one of the majors. - Page 3 line 73: I would start e.g., like "The main components of the risk management framework (Figure 1), namely ... - Page 3 lines 82-84: not all under "" is in italic. - Page 3 section 2.1: This section, which is almost completely composed by definitions, can be made as table, so that people familiar with the terminology can also decide to skip. - Page 4 line 99: Vulnerability not necessarily indicates a possible future state, but can indicate also a present state. - Page 4 line 109: you mention database development in relation to the creation of an archive (or inventory) of historical data and events. However, such archive do not necessarily means creating a database (although often it is a database). I would therefore use a more general term rather than a specific technological choice. - Page 4 line 114: the element GIS is not related to the previous ones in the list. - Page 4 line 118: You mention risk quantification, but after you switch to hazard. - Page 4 line 121: Often there is confusion between vulnerability and susceptibility. You can consider if give a definition of both to clarify which definition you are following. - Page 5 line 138: I would remove the acronym from the title and put after the first occurrence in the text. - Page 6 line 159: A DSS is not necessarily web based- - Page 6 line 169: Dalezios and Eslamian, 2026 -> 2016. - Page 6 line 176: In a previous section you say that Enviornmental hazards include also natural hazard and I would include also Meteological hazards into those, therefore I would use only of the two terms. - Page 7 line 190: Advances in environmental rely: maybe missing a word. - Page 7 line 190: In the following, : maybe missing a word. - Page 7 lines 191-191: maybe need to revise the entire sentence. - Page 7 line 206 up to 5 - up to five times the - Page 9 lines 261-262: The tornadoes and waterspouts are identical phenomena, the first definition is used over land and the second over sea. -> Tornadoes and waterspouts are the same phenomena, but the first occur over land while the second over sea. - Page 9 section 3.3: you should clarify if when you mention only tornados, you refer only to events over land or you refer to bot tornado and waterspouts. - Page 9 line 267: damage -> damages - Page 9 line 278: you have not given a definition of funnel clouds and why is different from tornados. - Page 9 line 284: Figure 5 -> Figure 6 - Page 10 line 288: was -> were - Page 10 line 304: Figure 6 -> Figure 7 - Page 10 line 316: Figure 7 -> Figure 8 - Page 11 lines 340-350: Several researchers have noted the importance of convection and especially, mesoscale convective systems in producing warm season precipitation. -> Several researchers have noted the importance of convection, and especially mesoscale convective systems, in producing warm season precipitation. - Page 12 line 352: using satellite imagery found -> by using satellite imagery was found - Page 12 line 367: forcing -> forcings - Page 12 line 379: Figure 8 - Figure 9 - Page 12 line 382: and covers a -> and covering a - Page 13 line 385: drought -> droughts - Page 13 lines 389-390: replace four cases of )( with ; - Page 13 line 404: Figure 9 -> Figure 10 - Page 13 line 411: Figure 10 -> Figure 11 - Page 14 line 442: Figure 11 -> Figure 12 - Page 14 line 446: analyze the results and case studies for each of the three steps of risk assessment: this does not emerge from the text. - Page 15 line 454: floods: this impact in not among the case studies.

- Figure 1: Above the top box is written Risk Assessment, which includes three elements, one of which is again Risk Assessment. - Figure 2: You have Hazard Assessment twice and can be a bit confusing. Further it is not clear why some elements that appear similar in the figure are considered in different ways or hierarchical levels in the text (e.g. under Risk Evaluation you only present DSS and they are useful fr other elements wich graphically are represented in the same way). - Figure 5: this figure is not cited in the text. - Figure 7: what do numbers indicate? - Figure 8: in the legend is showed tornadoes. Is that correct? - Figure 9: The caption of the graphs on the left seems not consistent with the text.
* * *
[Figure]

2020-155, 2020.

---

## Referee Comment (RC2) · Anonymous Referee #2 · 3 Jan 2021

This study provides a novel insights of risk management, that is the current risk assessment framework which consists of three steps (risk identification, risk estimation and risk evaluation) should complement the other procedure (the need for a feedback of all the risk assessment undertakings). According to the conception, it seems kind of interesting, however the full text may be too simple and more like a report to introduce different parts of risks. All figures are adapted from others' researches . It is so hard to accept, because these figures lack of originality. The authors should add more original figures. From the above reasons, the manuscript should be overhauled. In addition, I would like to give the authors some suggestions as follows. 1. The keywords listed by the author are scant, thus I suggest the authors add more keywords such as risk

management framework, Mediterranean ecosystems etc. 2. Lines 41. The authors emphasize the sustainability in the environmental status and agricultural production. However, the article involves less discussion and description of the impact of hazards on agricultural production in other part. Therefore, the sentence needs to be replaced. 3. Introduction should be reorganized. The knowledge gap is not clear and the innovative points of research need to be summarized. 4. The logic of the second section "risk management framework" needs to be changed. The settings of chapter are better to follow these 5 parts: risk identification, risk estimation, risk assessment, risk evaluation and risk governance. Also, the authors should illustrate the relationship between these 5 parts, not only just explain the functions and definitions of these 5 parts separately. 5. Lines 139. It may be a good way to use "Quantitative risk assessment (QRA)" as title, which is a duplicate of the subtitle. 6. The authors selected 7 case studies of meteorological and environmental hazards. However, the author needs to add the reasons and some literature ground. 7. I have not seen the application of the risk assessment framework mentioned by the author in these case studies. In other words, risk assessment framework, there is a lack of connection between the case studies and the risk framework. 8. The summary and discussion are too short. The author needs to summarize the deficiencies of existing research, improvement of risk management framework proposes future risk research prospects, especially for the Mediterranean ecosystems, according to case studies and others. 9. The resolutions of figures are difficult to read. Keep all the fonts in one format. The figures should be further revised to make some new insight comparing to the previous study the authors adapted from.

---

## Author Comment (AC1) · 19 Jan 2021

I. General comments (1)The paper presents a double review: on one side there is the presentation of a framework for risk management, while on the other there is a review of extreme meteorological hazards occurring mainly (but not only) in the wider are of Greece and Eastern Mediterranean. As for the first review, it would be important to control used terminology and show similarities/dissimilarities with most recent literature. Answer 1: The comment is considered and the introduction is thoroughly revised.

(2)Further, it would be important to describe the entire workflow in a more fluid way and less as a list of definitions. Answer 2: The introduction is revised to cover this

comment.

(3)As for the second review, it presents several recent findings for the considered hazard and case study area. However, this part is not connected with the previous one. It would be important to connect the two parts to make understand why and how they are connected. Answer 3: The two parts are connected after the revision.

II. Specific comments

(1)Page 2 line 35: I would not start the paper with that sentence, rather I would mention the several factors at the global scale and then say that environmental degradation is one of the majors. A1. It is revised. (2)Page 3 line 73: I would start e.g., like "The main components of the risk management framework (Figure 1), namely ... – Page 3 lines 82-84: not all under "" is in italic. A2. It is revised and corrected. (3)Page 3 section 2.1: This section, which is almost completely composed by definitions, can be made as table, so that people familiar with the terminology can also decide to skip. A3. This section was transferred into Table 1. (4)Page 4 line 99: Vulnerability not necessarily indicates a possible future state, but can indicate also a present state. A4. It is revised and corrected. (5)Page 4 line 109: you mention database development in relation to the creation of an archive (or inventory) of historical data and events. However, such archive do not necessarily means creating a database (although often it is a database). I would therefore use a more general term rather than a specific technological choice. A5. It is revised and corrected. (6)Page 4 line 114: the element GIS is not related to the previous ones in the list. A6. It is considered. (7)Page 4 line 118: You mention risk quantification, but after you switch to hazard. A7. It is considered. (8)Page 4 line 121: Often there is confusion between vulnerability and susceptibility. You can consider if give a definition of both to clarify which definition you are following. A8. It is revised and corrected. (9)Page 5 line 138: I would remove the acronym from the title and put after the first occurrence in the text. A9. It is corrected. (10)Page 6 line 159: A DSS is not necessarily web based. A10. It is revised and corrected. (11)Page 6 line 169: Dalezios and Eslamian, 2026 -> 2016. A11. It is corrected. (12)Page 6 line 176: In

a previous section you say that Environmental hazards include also natural hazard and I would include also Meteorological hazards into those, therefore I would use only of the two terms. A12. It is revised and corrected. (13)Page 7 line 190: Advances in environmental rely: maybe missing a word. A13. It is corrected. (14)Page 7 lines 191-191: maybe need to revise the entire sentence. A14. It is revised. (15)Page 7 line 206 up to 5 - up to five times the.. A15. It is corrected. (16)Page 9 lines 261-262: The tornadoes and waterspouts are identical phenomena, the first definition is used over land and the second over sea.  -> Tornadoes and waterspouts are the same phenomena, but the first occur over land while the second over sea. A16. It is revised and corrected. (17)Page 9 section 3.3: you should clarify if when you mention only tornados, you refer only to events over land or you refer to bot tornado and waterspouts. A17. It is revised and corrected. (18)Page 9 line 267: damage -> damages A18. It is corrected. (19)Page 9 line 278: you have not given a definition of funnel clouds and why is different from tornados. A19. Funnel cloud is defined and included. (20)Page 9 line 284: Figure 5 -> Figure 6 A20. Figure numbering is thoroughly considered. (21)Page 10 line 288: was -> were A21. It is corrected. (22)Page 10 line 304: Figure 6 -> Figure 7 A22. Figure numbering is thoroughly considered. (23)Page 10 line 316: Figure 7 -> Figure 8 A23. Figure numbering is thoroughly considered. (24)Page 11 lines 340-350: Several researchers have noted the importance of convection and especially, mesoscale convective systems in producing warm season precipitation. -> Several researchers have noted the importance of convection, and especially mesoscale convective systems, in producing warm season precipitation. A24. It is corrected. (25)Page 12 line 352: using satellite imagery found -> by using satellite imagery was found A25. It is corrected. (26)Page 12 line 367: forcing -> forcings A26. It is corrected. (27)Page 12 line 379: Figure 8 - Figure 9 A27. Figure numbering is thoroughly considered. (28)Page 12 line 382: and covers a -> and covering a A28. It is corrected. (29)Page 13 line 385: drought -> droughts A29. It is corrected. (30)Page 13 lines 389-390: replace four cases of )( with ; A30. It is corrected. (31)Page 13 line 404: Figure 9 -> Figure 10 A31. Figure numbering is thoroughly considered. (32)Page

13 line 411: Figure 10 -> Figure 11 A32. Figure numbering is thoroughly considered. (33)Page 14 line 442: Figure 11 -> Figure 12 A33. Figure numbering is thoroughly considered. (34)Page 14 line 446: analyze the results and case studies for each of the three steps of risk assessment: this does not emerge from the text. A34. It is revised. (35)Page 15 line 454: floods: this impact in not among the case studies. A35. It is revised and corrected. (36)Figure 1: Above the top box is written Risk Assessment, which includes three elements, one of which is again Risk Assessment. A36. It is corrected. (37)Figure 2: You have Hazard Assessment twice and can be a bit confusing. Further it is not clear why some elements that appear similar in the figure are considered in different ways or hierarchical levels in the text (e.g. under Risk Evaluation you only present DSS and they are useful fr other elements, which graphically are represented in the same way). A37. The flow-chart of risk management framework is thoroughly considered. (38)Figure 5: this figure is not cited in the text. A38. It is considered. (39)Figure 7: what do numbers indicate? A39. The numbers in the Figure (1-63) corresponds to the incidence of the 63 medicanes recorded from 1969-2014 as they reported in Nastos et al. (2015). (40)Figure 8: in the legend is showed tornadoes. Is that correct? A40. It is considered. (41)Figure 9: The caption of the graphs on the left seems not consistent with the text. A41. It is considered.

Please also note the supplement to this comment:
https://nhess.copernicus.org/preprints/nhess-2020-155/nhess-2020-155-AC1-supplement.pdf

---

## Author Comment (AC2) · 19 Jan 2021

I. General comments This study provides a novel insights of risk management, that is the current risk assessment framework which consists of three steps (risk identification, risk estimation and risk evaluation) should complement the other procedure (the need for a feedback of all the risk assessment undertakings). According to the conception, it seems kind of interesting, however the full text may be too simple and more like a report to introduce different parts of risks. All figures are adapted from others' researches. It is so hard to accept, because these figures lack of originality. The authors should add more original figures. From the above reasons, the manuscript should be overhauled.

[Figure]

Answer: The comments are thoroughly considered.

II. Specific comments

1. The keywords listed by the author are scant, thus I suggest the authors add more keywords such as risk management framework, Mediterranean ecosystems etc. A1. It is considered and revised. 2. Lines 41. The authors emphasize the sustainability in the environmental status and agricultural production. However, the article involves less discussion and description of the impact of hazards on agricultural production in other part. Therefore, the sentence needs to be replaced. A2. It is revised and corrected. 3. Introduction should be reorganized. The knowledge gap is not clear and the innovative points of research need to be summarized. A3. The Introduction has been revised. 4. The logic of the second section "risk management framework" needs to be changed. The settings of chapter are better to follow these 5 parts: risk identification, risk estimation, risk assessment, risk evaluation and risk governance. Also, the authors should illustrate the relationship between these 5 parts, not only just explain the functions and definitions of these 5 parts separately. A4. It is revised and corrected. The relationship between the 5 parts is illustrated through Figure 2. 5. Lines 139. It may be a good way to use "Quantitative risk assessment (QRA)" as title, which is a duplicate of the subtitle. A5. It is revised and corrected. 6. The authors selected 7 case studies of meteorological and environmental hazards. However, the author needs to add the reasons and some literature ground. A6. It is revised and corrected. 7. I have not seen the application of the risk assessment framework mentioned by the author in these case studies. In other words, risk assessment framework, there is a lack of connection between the case studies and the risk framework. A7. It is considered. These case studies fit mainly within risk identification and risk assessment. 8. The summary and discussion are too short. The author needs to summarize the deficiencies of existing research, improvement of risk management framework proposes future risk research prospects, especially for the Mediterranean ecosystems, according to case studies and others. A8. It is revised. 9. The resolutions of figures are difficult to read. Keep

all the fonts in one format. The figures should be further revised to make some new insight comparing to the previous study the authors adapted from. A9. The analysis of Figures is also considered.

Please also note the supplement to this comment:
https://nhess.copernicus.org/preprints/nhess-2020-155/nhess-2020-155-AC2-supplement.pdf

─────────────────────────